# Epidemiology and Genetic Diversity of Human Metapneumovirus in Patients with Severe Acute Respiratory Infection from 2023 to 2024 in Ningxia, China

**DOI:** 10.3390/diseases13080255

**Published:** 2025-08-11

**Authors:** Ting Mu, Jianxin Pei, Jingting Wang, Ling Niu, Zhonglan Wu

**Affiliations:** 1Ningxia Hui Autonomous Region Center for Disease Control and Prevention, Ningxia Hui Autonomous Region Academy of Preventive Medicine, NO.4 Fengchao Road, Yinchuan 750004, China; muting1209@126.com (T.M.); peijianxin282@163.com (J.P.); wjt569476979@126.com (J.W.); 2Zhongwei Center for Disease Control and Prevention, Zhongwei 755000, China; 18161593697@163.com

**Keywords:** human metapneumovirus, epidemiology, evolution, genotype

## Abstract

Background: Human metapneumovirus (HMPV) is a major pathogen responsible for causing severe acute respiratory infections (SARI). Whole-genome sequencing can better identify transmission events and outbreaks. In this study, we aimed to investigate the epidemiology and genetic diversity of HMPV in SARI cases in Ningxia, China. Methods: We collected respiratory tract samples from hospitalized patients with SARI from October 2023 to September 2024 in Ningxia, China. Nasopharyngeal swabs were tested for respiratory viruses with qRT-PCR. Whole-genome sequences were determined for samples with high viral loads using an amplicon-based method. Results: We enrolled 2873 SARI patients from October 2023 to September 2024, and found an HMPV-positive proportion of 3.06% (88/2873). Children aged 4 years were particularly susceptible to HMPV infection, with a positive proportion of 10.92% (13/119). HMPV exhibits distinct seasonal characteristics, consistent with its established epidemiological pattern, with a peak incidence occurring during winter months. Sixteen complete HMPV genome sequences were obtained. Among these, 81.25% (13/16) were identified as genotype A (A2.2.2: 92.31%, 12/13; A2.2.1: 7.69%, 1/13) and 18.75% (3/16) as genotype B1. Notably, the dominant strain was 111nt-dup in genotype A2.2.2. Sequence analysis of HMPV genes revealed divergent G-gene sequence identities between different genotypes. Additionally, the potential glycosylation sites of the G protein varied across genotypes. Conclusions: In this study, we found that the 111nt-dup strain was the dominant one in genotype A, and multiple genotypes co-circulated in Ningxia from October 2023 to September 2024. The HMPV G protein exhibited the highest level of inter-strain diversity between genotypes. These findings provide valuable insights into the prevention and control of HMPV infections in China.

## 1. Introduction

Human metapneumovirus (HMPV), first discovered in 2001, is a common cause of respiratory tract infections in children and a major concern for elderly and immunocompromised patients [1]. Serological studies have shown that HMPV has circulated in the human population for at least 60 years [2]. In 2018, it was estimated that approximately 11.1 million cases of acute lower respiratory infections (ALRI) globally were attributed to HMPV infections, leading to 502,000 hospital admissions and 113,000 deaths [3]. National surveillance data confirmed that HMPV was responsible for 6.2% of respiratory illnesses and 5.4% of hospitalizations in China during December 2024, with multiple regions, including China (incorporating Hong Kong), Kazakhstan, and Malaysia, reporting marked increases in cases by early January 2025 [4]. A prospective study indicated that the risk of severe disease caused by HMPV in infants under one year of age is comparable to that of RSV and influenza, and 79% of HMPV-related deaths occurring globally in infants aged 0–5 months in low- and middle-income countries [3]. Notably, despite the higher prevalence of influenza viruses, comparative studies suggested that HMPV and RSV pose a higher risk of severity in hospitalized adult cases [5]. The clinical manifestations of HMPV infection range from a mild upper respiratory tract infection to life-threatening severe bronchitis and pneumonia [6]. Currently, there are neither specific drugs nor licensed vaccines against HMPV infections [7]. Some research is exploring new strategies and drugs to treat HMPV infections.

HMPV is a single-stranded, negative-sense, nonsegmented RNA virus that was reclassified from the family *Paramyxoviridae* to the family *Pneumoviridae* in 2016 [8]. The genome organization of HMPV is quite similar to that of avian metapneumovirus (AMPV) [8]. The HMPV genome is approximately 13.3 kb and contains eight genes that code for nine proteins. The fusion (F) and attachment (G) proteins are two major surface glycoproteins of HMPV, which play important roles in viral replication and host immune response. Based on the antigenic properties of the G and F protein, HMPV is classified into two genotypes, A and B, and is further subdivided into genotypes such as A1, A2a, A2b, A2c, B1, and B2 [9,10,11,12]. In addition, HMPV strains with 180 and 111 nucleotide duplication (nt-dup) in the G gene have been identified in Japan since 2014 and 2017, respectively [13,14]. All these novel variants were scattered in a cluster of A2c lineage and designated as A2c_180nt-dup_ and A2c_111nt-dup_ strains, respectively. This indicates that these variants were likely the result of multiple incidences of mutations rather than having evolved from a single mutation strain. Recently, variants of HMPV G gene with 180 nt-dup or 111 nt-dup have been detected across multiple regions, including Japan, Spain, and the Netherlands [15,16,17]. Since 2016, these variants have gradually replaced other A genotypes to become the globally dominant A2c lineage [18]. Epidemiological surveillance data from China indicated that the A2c_111nt-dup_ variant is the predominant strain in Henan (2017–2019) and Nanjing (2021–2022). Although other genotypes, including A2b, B1 and B2, were detected, their prevalence remained substantially lower [19,20].

The unique climatic and multi-ethnic demographic context may shape distinctive HMPV evolutionary ecology in Ningxia. Regional studies have revealed a significantly higher prevalence in Ningxia, where HMPV accounted for >10% of hospitalized pediatric acute respiratory infection cases during the 2011–2012 and 2018–2019 surveillance periods, with coinfections occurring in >50% of these cases [21,22]. Molecular epidemiology confirmed that both 2021 Ningxia strains belong to genotype B2, with high sequence identities to Australian 2020 strains, suggesting cross-border transmission [23]. Due to limited samples and restricted temporal scope, current research cannot fully elucidate HMPV diversity landscape or its evolutionary drivers in Ningxia. Here, we conducted a study on HMPV in SARI cases in Ningxia from October 2023 to September 2024, aiming to explore its epidemiological trends and genetic diversity and provide support for the implementation of control measures.

## 2. Methods

### 2.1. Case Sources and Sample Collection

From October 2023 to September 2024, nasopharyngeal swabs were collected from cases of severe acute respiratory infections (SARI) in five prefecture-level influenza surveillance sentinel hospitals distributed throughout the Ningxia region. Based on the National Sentinel Surveillance Technical Program for Acute Respiratory Infectious Diseases (Trial Version), SARI cases in this study were defined as those admitted to the hospital or within 48 h of admission with acute onset, a history of fever (measured fever of ≥38 °C) and cough, and where the onset did not exceed 10 days. Participants in this study ranged in age from 7 days to 98 years. All samples were stored in 3 mL of viral transport medium (VTM) and kept at −80 °C for further processing. Primary screening was performed in local laboratories using standardized multiplex PCR kits. HMPV-positive samples were stored at −80 °C and transported to the Ningxia CDC laboratory in time for review and whole-genome sequencing.

### 2.2. Detection of Human Metapneumovirus

Total viral nucleic acids were extracted from 200 µL of specimens and eluted with 50 µL of water using a magnetic-bead-based extraction kit (Xi’an Tianlong Science & Technology Co., Ltd., Xi’an, China) according to the manufacturer’s instructions. A one-step multiplex real-time reverse transcription polymerase chain reaction (qRT-PCR) method (Shanghai Biogerm Medical Technology Co., Ltd., Shanghai, China, LOT: SJ-HX-807-2) was employed to detect six respiratory viruses: adenovirus (AdV), respiratory syncytial virus (RSV), parainfluenza virus 1 (PIV1), parainfluenza virus 3 (PIV3), human rhinovirus (HRV), and HMPV. The experimental methods and results were interpreted in accordance with the kit instructions. The detection limit of the method was 10^3^ copies/mL and a positive result was indicated by a CT value of less than 35.

### 2.3. Epidemiological Method

Epidemiological analysis was performed on laboratory-confirmed HMPV cases using standardized case report forms to collect demographic characteristics (age, sex, and residence) and temporal variables (date of sampling and testing). Age groups were categorized according to the age distribution of cases in the National Influenza Surveillance System (0–4 years, 5–14 years, 15–44 years, 45–64 years, and ≥65 years). Statistical methods were then applied to analyze the data.

### 2.4. Whole-Genome Sequencing

HMPV-positive samples with high viral loads (Ct < 30) were further used for whole-genome sequencing by an amplicon-based method. Viral nucleic acid was amplified by a HiScript^®^II one-step RT-PCR kit (Vazyme Biotech Co., Ltd., Nanjing, China, LOT: P611-01) using four pairs of overlapping primers that were used as previously described [24]. The HMPV genomic amplicons were purified using magnetic beads (Beijing MicroFuture Technology Co., Ltd., Beijing, China, LOT: MF63880) according to the manufacturer’s instructions. The purified DNA was then quantified and diluted to 0.5 ng/μL with a Qubit^TM^ dsDNA HS Assay Kit (Invitrogen, Carlsbad, CA, USA, LOT: 2489015). Thereafter, the MFEasy^®^ Library Kit (Beijing MicroFuture Technology Co., Ltd., Beijing, China, LOT: MF180226) was employed to create a library following the instructions. The purification and quantification of the library were repeated following the above operations to precisely adjust the final concentration to 1.1 ng/μL and combine all the samples to create a single library. After denaturation, sequencing was carried out on the Illumina MiSeq instrumentusing the Miseq^®^ v2 300 cycle Kit (Illumina, San Diego, CA, USA).

### 2.5. Bio-Informatics Analysis

Raw sequences were assembled de novo using CLC Genomics Workbench 23.0.1, and finally a majority consensus genome was extracted. Phylogenetic analysis was performed by aligning all sequences generated in this study against the representative references obtained from NCBI using MEGA software version X (Mega Ltd., Auckland, New Zealand). The maximum likelihood method was utilized to construct the phylogenetic tree with 1000 bootstrap replicates. The GTR + G + I model was found to be the best DNA/protein model for nucleotide sequences, while the JTT + G + I model was found to be the best model for amino acid sequences, both of which were used for the HMPV evolutionary analysis. The sequence identities of nucleotide sequences and amino acid residues encoded in the whole-genome sequences were calculated using Megalign 7.1.0.44 (DNASTAR). The positive residues of N-linked and O-linked glycosylation were predicted using the online websites of NetNGlyc-1.0 (https://services.healthtech.dtu.dk/services/NetNGlyc-1.0/) and NetOGlyc-4.0 (https://services.healthtech.dtu.dk/services/NetOGlyc-4.0/, respectively, accessed on 3 December 2024).

The reference sequences used in this study were obtained from the NCBI database. Considering that not all published sequences were applicable, we selected 27 reference sequences from different countries around the world, such as China, the Netherlands, and the United States, for analysis based on the genetic differences between the strains, their geographical origin, and the availability of high-quality complete sequences. 

### 2.6. Statistical Analysis

Data analysis was performed using SPSS 27.0, and categorical data were compared using the Chi-square (χ^2^) test. A two-tailed *p*-value of <0.05 was considered statistically significant.

## 3. Results

### 3.1. Epidemiology of HMPV Infection

A total of 2873 SARI cases were enrolled in Ningxia from October 2023 to September 2024. The results of qRT-PCR revealed that the HMPV-positive proportion was 3.06% (88/2873). As shown in Table 1, the positive proportion in males was slightly lower than that in females, though no statistically significant gender difference was observed (χ^2^ = 0.161, *p* = 0.688). The age of the enrolled patients ranged from 7 days to 98 years, with a significantly different distribution of HMPV prevalence across age groups. The highest HMPV-positive proportion (5.58%, 25/448) was observed in children under 5 years old (χ^2^ = 22.298, *p* < 0.001). Within this group, the highest positive proportion (10.92%, 13/119) was found in 4-year-old children (χ^2^ = 10.595, *p* = 0.014). The seasonal distribution of HMPV infection from October 2023 to September 2024 showed that HMPV mainly concentrated from October to March, peaking in winter and spring, and nearly disappearing in the summer and autumn. The highest positive proportion of HMPV occurred in December (12.02%, 22/183), indicating a substantial increase in comparison to other months (χ^2^ = 121.039, *p* < 0.001) (Figure 1).

### 3.2. Coinfection with Other Respiratory Viruses

Among the 88 HMPV-positive SARI cases, mono-infections were predominant (68/88, 77.3%) (Table 2). A total of 20 coinfections (22.7%) were detected; coinfection of HMPV with Human rhinovirus (HRV) or Mycoplasma pneumoniae (MP) was most common (6/20, 30%), followed by SARS-CoV-2 (3/20, 15%); and respiratory syncytial virus (RSV) (3/20, 15%) and Adenovirus (AdV) exhibited the lowest coinfection rates (2/20, 10%).

### 3.3. Phylogenetic Analysis of HMPV Whole Genome

To understand the lineage distribution and epidemiological causes of HMPV in Ningxia, we analyzed 16 whole-genome sequences of HMPV from this study, along with published HMPV sequences, to determine their genotype using a phylogenetic approach (Figure 2). Among the samples analyzed, 81.25% (13/16) were identified as genotype HMPV-A, while 18.75% (3/16) were classified as HMPV-B. Under the novel lineage nomenclature, phylogenetic clusters are designated as follows: A1, A2, A2.1, A2.2, A2.2.1, A2.2.2, B1, and B2 [16]. According to the phylogenetic analysis, the sequences identified in this study could be divided into the following genotypes: A2.2.2 (75%, 12/16), A2.2.1 (6.25%, 1/16), and B1 (18.75%, 3/16). Among the A2.2.2 strains, 91.67% (11/12) of the sequences exhibited nucleotide duplication (nt-dup) in the G gene, and all were classified as the 111nt-dup variant.

Since the genes of HMPV are consistently located in a linear genome, phylogenetic incongruence among individual HMPV genes may suggest past recombination events [10]. In this study, nucleotide sequence analysis revealed no changes in the phylogenetic positions of these eight genes across lineages. However, amino acid sequence analysis showed that some strains exhibited different phylogenetic positions for the M and N genes. The B1 M sequences were located within the B2 genotype, while some M and N sequences from A2.2.2 strains were assigned to the A2.1 or A2.2.1 genotype (Figure 3). The incongruence between nucleotide- and amino acid-based phylogenetic reconstructions indicates that complex evolutionary events in HMPV are not fully resolved by nucleotide sequence analysis alone.

## 4. Sequence Analysis of HMPV Genes

Significant genetic differences were observed between the HMPV genotypes A and B, with an inter-genotype nucleotide sequence identity of less than 81.9%. Within genotype A, the A1 lineage was distinct from all A2 lineages. However, similar sequence identities were observed among A2.1, A2.2.1, and A2.2.2. Intra-genotype homology was consistently high, with nucleotide sequence identities ranging from 94.1% to 100% within genotype A, from 98.9% to 99.9% within genotype B, and from 98.1% to 100% within the 111nt-dup sequences.

A comparative analysis of individual genes (see Table 3) revealed that the nucleotide and amino acid sequences of the N, P, M, F, M2, and L genes were highly conserved between genotypes, averaging above 80%, with even greater conservation observed within each genotype, averaging above 95%. However, significant differences were observed in the nucleotide and amino acid sequences of the SH and G genes between the two genotypes. The G gene exhibits relatively lower nucleotide and amino acid sequence identity.

## 5. Conserved and Divergent Features of HMPV G Protein

As shown in the results above, the G protein of HMPV is the most divergent structural protein among the strains, being highly variable and glycosylated. Online predictions indicated that the HMPV G protein contains two to five potential N-linked glycosylation sites and more than 60 potential O-linked glycosylation sites. The N-linked glycosylation sites at aa30 and aa52 were conserved in the G sequences of genotype A (Table 4). For other N-linked glycosylation sites, intra-genotypic variations were relatively minor, but substantial differences were observed between genotypes A and B. Similarly, serine and threonine, which serve as potential O-linked glycosylation receptor sites, were present in all virus strains but varied greatly. The number of serine residues ranged from 17 to 43, with genotype A strains having more serine residues than genotype B (Table 4). In contrast, the number of threonine residues ranged from 33 to 51, with genotype B strains containing more threonine residues. Due to the insertion of 111 nucleotides in the A2.2.2 variant, it has significantly more O-linked glycosylation sites than other genotypes. This diversity of amino acid sequences along with the disparities in glycosylation sites probably aids in its evasion of the immune system [25].

## 6. Discussion

HMPV has been identified as one of the pathogens of acute respiratory tract infections (ARI) since it was discovered in 2001 [2]. The Centers for Disease Control and Prevention (CDC) identifies transmission primarily through respiratory secretions, close contact, and contaminated surfaces or fomites [26]. Almost all children are eventually infected with HMPV, and adults may be reinfected with HMPV throughout their lives due to incomplete immunity [27]. Previous studies have shown that the prevalence of HMPV among inpatients with ARI is approximately 6% worldwide [28]. A multicenter prospective study of 2733 hospitalized children with ARI and HMPV was detected in 5.3% of the respiratory specimens from 2017 to 2019 [29]. Early warnings of HMPV outbreaks in China in December 2024 and the subsequent surge in cases across Malaysia, Kazakhstan, and other parts of Asia in January 2025 underscore the virus’s growing public health importance [4]. 

To better understand the epidemiological trends and genetic characteristics of HMPV in Ningxia, we conducted a study in cases of SARI from October 2023 to September 2024. The overall positive proportion for HMPV was 3.06%, which was comparable to report rates in other Chinese regions, including Beijing (4.08%) [30], Nanjing (4.7%) [19], and Huzhou (4.94%) [31]. There was no significant gender difference in HMPV infections. The majority of HMPV cases occurred in children under five years old, especially those aged four, a finding that aligns with the majority of prior studies [30,31]. Global surveillance data indicate that HMPV typically reaches its peak prevalence during the winter and early spring [32]. In both northern and southern regions in China, HMPV was primarily detected between March and May [18,33]. This study confirmed the winter and spring seasonal characteristics of HMPV in Ningxia, with a peak in December and a positive proportion of 12.02%. Based on these results, local public health authorities can strengthen health promotion and reserve medical resources before an outbreak occurs. The focus should be on protecting young children, teaching proper hand-washing methods, developing good hygiene habits, emphasizing the wearing of masks in crowds, and maintaining social distance to reduce the spread of HMPV.

Coinfection of HMPV with other respiratory pathogens, such as RSV, HRV, coronavirus (CoV), influenza virus (IFV), and parainfluenza virus (PIV), has been shown in several studies and may lead to complicated clinical outcomes [29,34]. Among the 88 HMPV-positive SARI cases included in this study, the coinfection proportion was 22.7% (20/88), primarily involving coinfection with HRV or MP (30%, 6/20), followed by SARS-CoV-2, RSV, and AdV. The clinical significance of coinfection remains controversial, and current evidence suggests that the association between HMPV coinfection and intensive care unit (ICU) admission rates or disease severity is unclear [1,35]. Further research is needed to validate this hypothesis, which will involve analyzing large-scale surveillance data. Continuous monitoring and determination of the epidemiological characteristics of HMPV in Ningxia is crucial for regional prevention and control. Furthermore, the advancement of continuous epidemiological monitoring of HMPV at the national and global levels is equally significant for the formulation of precise public health policies.

The whole-genome sequences of 16 HMPV strains obtained in this study revealed the co-circulation of three genotypes, among which the 111nt-dup in genotype A2.2.2 was dominant, which is consistent with the results of HMPV studies in China in recent years [36]. Research has indicated that 111nt-dup has emerged as the predominant genotype of HMPV in various regions, including Japan, Spain, and Nanjing in China [14,15,19]. A total of 11 sequences with 111nt-dup variants were detected in this study, indicating ongoing geographical spread in China. Our data showed that genotype proportions generally align with global trends. The low detection rate of genotype B may reflect the true epidemiological pattern in Ningxia from 2023 to 2024. However, due to the limited sample size, we cannot rule out the possibility that minor genotype B variants may have been underestimated. A separate analysis of HMPV genes showed that the G gene was the most prone to mutations, followed by the SH gene, while the F, N, P, M, M2, and L genes were highly conserved. In addition, phylogenetic analysis showed that in amino acid-based trees, the position of the N genes of some strains was replaced by genotype A2.1 instead of A2.2.2, and the M genes of some strains were more inclined to belong to genotype B2 rather than B1. Notably, these discrepancies were absent in nucleotide-based trees. These findings indicated that phylogenetic analysis of key structural proteins (e.g., M and N gene) at the amino acid level may reveal cryptic evolutionary events, even in the absence of apparent recombination signals in the M and N genes of HMPV (bootscan values < 70%). However, it should be noted that the study of HMPV recombination remains an emerging field requiring further validation. Future studies could employ larger sample cohorts and long-read sequencing technologies to further investigate potential recombination mechanisms in HMPV. The concurrent discordant phylogenetic placements of both M and N genes within the same viral strain, which deviated from their phylogenetic backbone and pointed to different genotypes, suggested potential recombination events [10]. This necessitates integrated multi-level and multi-gene analysis strategies in viral phylogenetics, and requires whole-genome scanning of anomalous strains for recombination using tools such as RDP5 and SimPlot, thereby elucidating HMPV evolutionary dynamics.

HMPV G protein is predicted to be a type-II transmembrane protein that may contribute to viral attachment [37]. However, the function of HMPV G protein remains unclear. We attempted to find critical functional sites by analyzing the changes in the amino acid sequence of the G protein. Glycosylation is an important modification of proteins, which plays a crucial role in regulating proteins and maintaining protein stability. Bioinformatic predictions revealed significant differences in the N-linked and O-linked glycosylation sites of HMPV G protein among different genotypes, suggesting that genotype-dependent glycosylation patterns may be a key driver of dominant strain prevalence. Notably, the substantial duplication sequences in the dominant 111nt-dup variant increased the potential glycosylation sites and extended the extracellular domain of the G protein [38], suggesting that glycosylation patterns may be a key factor driving the adaptive evolution of epidemic strains. This finding is consistent with the report by Van den Hoogen et al., which indicated that the number of N-linked sites in HMPV G protein range from two to six, with only one site (aa30) being strictly conserved [39]. However, as this site is located at the junction of the cytoplasmic tail and the transmembrane structural domain, the likelihood of actual N-linked glycosylation is low [25]. Furthermore, the spatial relationship between glycosylation and antigenic epitopes of HMPV G protein remains unclear. Amino acid mutations may mediate immune escape by altering epitope conformation [40,41], but the precise antigenic determinants of the G protein have not yet been identified. Therefore, it is urgently necessary to validate whether glycosylation sites are located in functional antigenic domains through structural biology and immunological cross experiments, to elucidate the regulatory mechanisms of protein modifications on immunogenicity.

This study has some limitations. Since all samples included in this study were obtained from SARI cases, it is still uncertain whether there is an association between the severity of HMPV infection and specific genotypes, particularly given our lack of standardized clinical severity assessment. This limitation precludes meaningful investigation of genotype–phenotype correlations (e.g., the 111nt-dup variant) and their potential clinical implications. Given the limited duration of continuous pathogen monitoring of ARI in Ningxia, and the fact that only 18.18% (16/88) of HMPV-positive cases had whole-genome sequences available, the conclusions regarding the epidemiological situation and genotypes of HMPV may be biased. Whether the transformation of its major dominant genotype follows a certain regular periodic pattern also requires longer-term surveillance. Future studies incorporating standardized severity assessments, expanded genomic surveillance, and longer observation periods will be crucial for elucidating the clinical significance of genetic variations and establishing robust epidemiological patterns of HMPV circulation.

A further limitation of this study is that the detection pathogen did not include the four common human coronaviruses (229E, NL63, OC43, and HKU1), which are recognized as aetiologic agents of respiratory infections, particularly in pediatrics and elderly populations. While the findings of our other parallel study indicate that these coronaviruses may have contributed relatively less to the overall disease burden in our setting in comparison to other common viruses (unpublished data), this omission may nevertheless have resulted in an underestimation of coinfection rates and an incomplete characterization of the full spectrum of circulating respiratory viruses.

## 7. Conclusions

In conclusion, this study has provisionally revealed the epidemiological patterns and genetic diversity of HMPV in Ningxia, China. This information has important reference value for public health authorities in formulating HMPV prevention and control strategies.

## Figures and Tables

**Figure 1 diseases-13-00255-f001:**
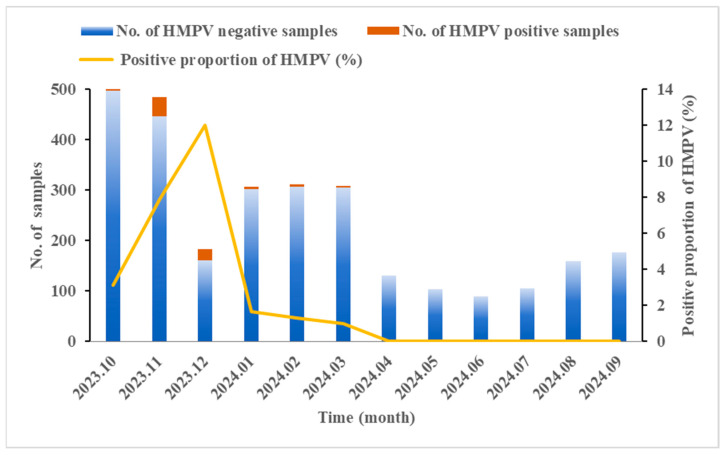
Seasonal distribution of HMPV in SARI cases from October 2023 to September 2024 in Ningxia, China. The Chi-square test revealed significant monthly variation in the HMPV-positive proportion (χ^2^ = 121.039, *p* < 0.001).

**Figure 2 diseases-13-00255-f002:**
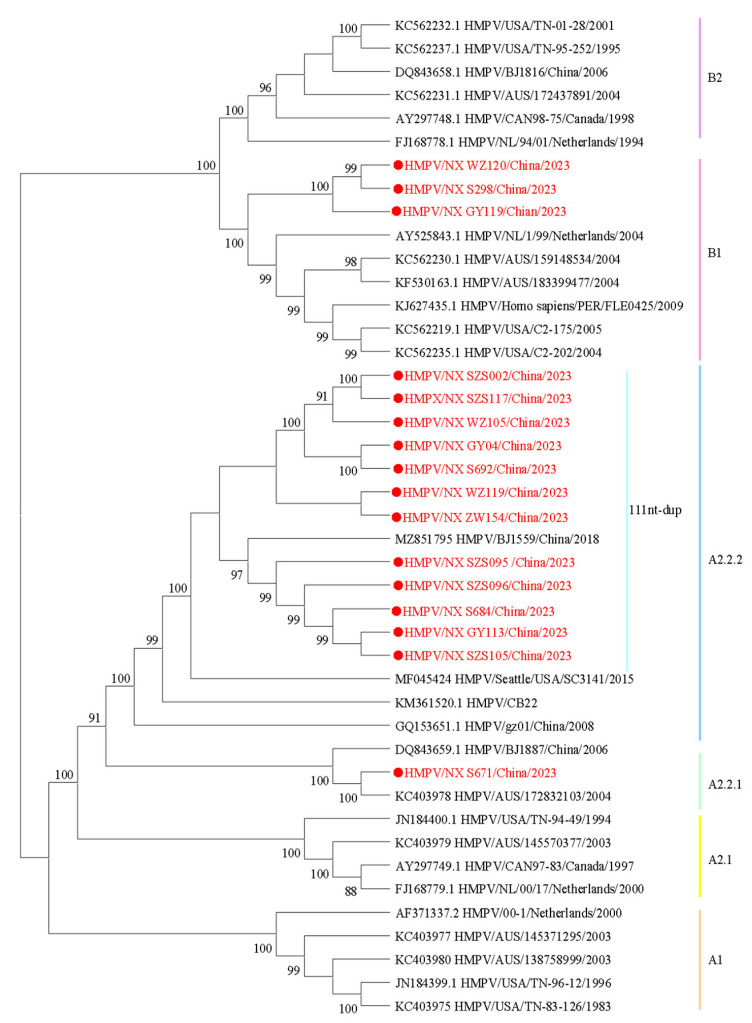
Phylogenetic tree constructed based on the HMPV whole-genome sequences using the maximum likelihood method with 1000 replicates. Sequences in this study are marked by ● with red, and the representative strains are shown in black. Different colors represent different genotypes. The bootstrap values (>80%) are shown above the branches.

**Figure 3 diseases-13-00255-f003:**
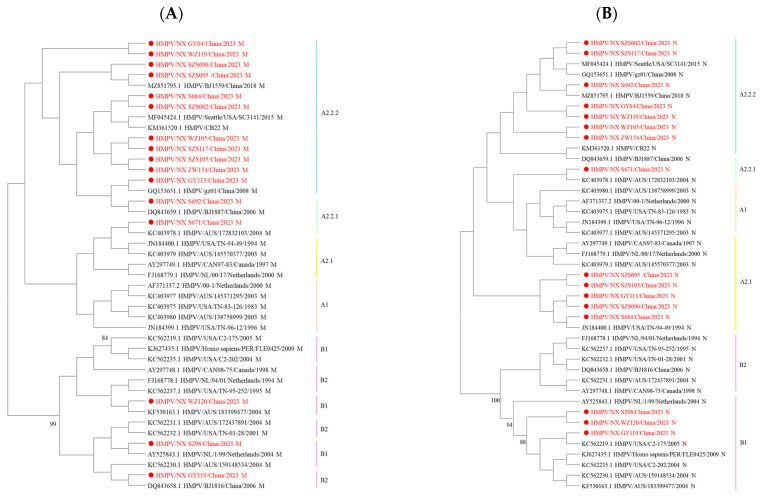
Phylogenetic tree of 16 HMPV strains was generated using the maximum likelihood method based on the *JTT* model with 1000 replicates. (**A**) Phylogenetic tree based on the HMPV M amino acid sequences. (**B**) Phylogenetic tree based on the HMPV N amino acid sequences. Sequences in this study are marked by ● with red, and the representative strains are shown in black. Different colors represent different genotypes. The bootstrap values (>80%) are shown above the branches.

**Table 1 diseases-13-00255-t001:** HMPV-positive proportion in SARI cases of different gender and ages from October 2023 to September 2024 in Ningxia, China.

Variable	Number of Samples Tested	Number of HMPV-Positive Samples	Positive Proportion of HMPV (%)	*χ* ^2^	*p*-Value
Gender				0.161	0.688
Male	1497	44	2.94		
Female	1376	44	3.20		
Age (years group)				22.298	*p* < 0.001
0–4	448	25	5.58		
~1	173	9	5.20	10.595	0.014
~2	68	1	1.47		
~3	88	2	2.27		
~4	119	13	10.92		
5–14	610	28	4.59		
15–44	298	5	1.68		
45–64	459	9	1.96		
≥65	1058	21	1.98		
Total	2873	88	3.06		

Abbreviation: HMPV = human metapneumovirus.

**Table 2 diseases-13-00255-t002:** Coinfection of HMPV and other respiratory pathogens detected in the study.

	Viruses Detected in Coinfection	No. (%, *n* = 88)
Single infection		68 (77.27)
Coinfection	HMPV + SARS-CoV-2	3 (3.41)
	HMPV + RSV	3 (3.41)
	HMPV + HRV	6 (6.82)
	HMPV + AdV	2 (2.27)
	HMPV + MP	6 (6.82)
Total		88

**Table 3 diseases-13-00255-t003:** Identity of nucleotide and amino acid sequences within and between HMPV genotypes in this study.

Gene	% Nucleotide Sequence Identity Within Genotype		% Nucleotide Sequence Identity Between Genotype A and B	% Amino Acid Sequence Identity Within Genotype		% Amino Acid Sequence Identity Between Genotype A and B
	A	B		A	B	
N	95.9–100	98.7–99.9	85–86.9	99.2–100	98.7–100	93.9–95.7
P	96–100	99.9–100	79.2–80.5	96.6–100	100	83.1–84.7
M	95.6–100	99.7–99.9	84.2–86.3	99.6–100	100	97.3–97.6
F	95.1–100	99.9–100	83.3–84	99.1–100	99.8–100	94.1–94.6
M2	96.1–100	99.7–100	84.3–85.3	95.5–100	99.6–100	86–87.2
SH	94.2–100	98.7–99.6	68.8–71.5	92.4–100	98.4–100	57.8–61.6
G	73.9–100	99.5–99.6	52.7–55.4	64.7–100	99.3–99.8	30.5–37.8
L	95.8–100	98.3–99.9	83.9–85.7	98.8–100	99.3–99.8	93.8–94.7

**Table 4 diseases-13-00255-t004:** Predicted N-linked and O-linked glycosylation-positive residues of HMPV G protein in this study.

Genotype	Name	Length	N-Linked Glycosylation-		O-Linked Glycosylation (Number)	
			Number	Positive Residues	Serine	Threonine
A2.2.1	HMPV/NX S671/China/2023	228	5	30/52/140/145/155	34	42
A2.2.2	HMPV/NX SZS095/China/2023	220	3	30/52/145	34	33
111nt-dup	HMPV/NX GY04/China/2023	264	3	30/52/256	40	44
	HMPV/NX GY113/China/2023	264	5	30/52/122/145/256	43	44
	HMPV/NX S684/China/2023	264	5	30/52/122/145/256	43	43
	HMPV/NX S692/China/2023	264	3	30/52/256	42	44
	HMPV/NX SZS002/China/2023	264	3	30/52/256	40	45
	HMPV/NX SZS096/China/2023	264	5	30/52/122/145/256	42	42
	HMPV/NX WZ119/China/2023	264	5	30/52/152/218/256	43	45
	HMPV/NX WZ105/China/2023	264	3	30/52/256	42	46
	HMPV/NX SZS105/China/2023	264	5	30/52/122/145/256	43	44
	HMPV/NX ZW154/China/2023	265	4	30/52/138/256	39	46
	HMPV/NX SZS117/China/2023	264	3	30/52/256	40	45
B1	HMPV/NX GY119/China/2023	241	5	118/167/181/184/188	17	51
	HMPV/NX S298/China/2023	241	5	118/167/181/184/188	17	50
	HMPV/NX WZ120/China/2023	241	5	118/167/181/184/188	17	50

## Data Availability

The de-identified data were obtained from the Influenza Surveillance Network Laboratory and the NCBI.

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
