# Peer review of "Epidemiology and Genetic Diversity of Human Metapneumovirus in Patients with Severe Acute Respiratory Infection from 2023 to 2024 in Ningxia, China"

_diseases, 2025, doi:10.3390/diseases13080255_

Round 1

Reviewer 1 Report

Comments and Suggestions for Authors

This study determined the epidemiology and characterized the genetic diversity of human metapneumovirus in patients with severe acute respiratory infection between October 2023 and September 2024 in Ningxia, China. The findings reported that the positive rate of HMPV detected in SARI patients was 3.06% for all ages and 10.92% for children aged under 4 years. Additionally, the positive proportion of HMPV was predominant in winter and spring, with a peak occurring in December. Genetic analysis revealed that the circulating HMPV genotypes (n=16) during the study period belonged to genotypes A2 and B1. The predominant strain was A2c111nt-dup.

 1.     Remove the highlights section or summarize them in the discussion part. Please ensure to follow the journal template.

2.     To be clearer, please mention the study period in the title.

3.     Is HMPV a major pathogen causing severe respiratory infection? A previous study estimated the rate of HMPV infection among hospitalized cases with acute respiratory infection (ARI) to be only 6.24% (Lefebvre A et al., 2016, DOI: 10.1016/j.jcv.2016.05.015). Please clarify.

4.     Please check the definition of "SARI" used in this study. In reference 23, the severe acute respiratory infection was defined as "sudden onset of fever above 38°C, cough or sore throat, shortness of breath or difficulty breathing, and requiring hospitalization for individuals aged above five years. For individuals aged between two months and five years, the criteria used were either those for pneumonia in this age group, i.e., cough or difficulty breathing and a breathing rate above 60, 50, and 40 breaths per minute for those younger than two months, between two and 12 months, and older than 12 months, respectively, or those for severe pneumonia in this age group, i.e., cough or difficulty breathing, requiring hospitalization and showing at least one of five danger signs.

Reviewer 2 Report

Comments and Suggestions for Authors

This study systematically analyzes the epidemiological characteristics and genetic diversity of human metapneumovirus (HMPV) in patients with severe acute respiratory infection (SARI) in Ningxia, China, from 2023 to 2024. It fills an important regional data gap and holds considerable scientific and public health value.

Two minor issues:

  1. The terms “positive proportion” and “positive rate” are used interchangeably, but their meanings are not entirely equivalent. Please consider standardizing terminology throughout the manuscript.

  2. In Figure 1, is there a statistically significant difference between the peak month and other months? This should be clarified.

Reviewer 3 Report

Comments and Suggestions for Authors

The specific study represents a valuable addition to the epidemiology of human metapneumovirus. It is generally scientifically sound and well presented. hMPV and respiratory viruses in general pose a significant burden on public health and human society in general and molecular epidemiology studies of such important viruses, like hMPV, must always be welcome. A variety of the most modern molecular methods that aided to the genetic characterisation of medically important hMPV strains, including whole-genome sequencing and phylogenetic add substantial value to the significance of this work regarding the genetic epidemiology and evolution of hMPV.

In my opinion, the present study is suitable for publication, provided that some revisions are made first. Specifically, since the authors report that the discrepancies observed in the phylogenetic relationships on the strains between the M and G genes, perhaps their work should benefit from appropriate genetic recombinaiton analysis using a freely available software, such as SimPlot. Secondly, the authors sequenced only 16 out of the 88 hMPV-positive samples. If that is due to viral load limitations, most likely indicated by the Ct values of the Real Time PCR used for routine detection, they could state it. If there was another limitation (e.g. restricted resources), I would personally understand and I would definitely not consider that as a reason for not accepting the manuscript by any means. Finally, the authors do not provide any information about whether they have submitted their sequences to GenBank. And that is my only concern from an ethical point of view regarding the study, i.e. the sequence data should be publicly available.

Reviewer 4 Report

Comments and Suggestions for Authors

This study presents valuable data on human metapneumovirus (HMPV) among patients with severe acute respiratory infection (SARI) in Ningxia, China, during 2023–2024. The manuscript is well-structured and scientifically sound, providing important insights into seasonal trends, age-related susceptibility, genotype prevalence, and genetic characteristics of circulating HMPV strains. It makes a valuable contribution to the field of respiratory virus epidemiology and surveillance.

Comments.
1. Sample collection and reproducibility:
The methodology would benefit from further clarification regarding the sample processing step. Specifically, please indicate the volume of viral transport medium (VTM) used for storing nasopharyngeal swabs. This information is important for reproducibility, especially since the viral nucleic acid extraction protocol specifies both sample uses and elution volumes.

2. Age categorisation:
The age grouping used in this study follows the Chinese national guideline, which is appropriate. However, it should be noted that the age group 15–44 years includes adolescents (15–17 years) who are still in school and not yet of working age. Their exposure environment may differ significantly from adults, potentially affecting infection risk. 
Did you consider further sub-grouping within this range? 
A brief justification for the current grouping would be helpful.

3. Lack of data on human coronaviruses:
The viral detection panel used in this study did not include common human coronaviruses (229E, NL63, OC43, and HKU1), which are known aetiologic agents of respiratory infections and may cause SARI, especially in children and the elderly. 
This may lead to an underestimation of co-infection rates. 
I recommend acknowledging this omission as the Limitations in the Discussion section.

4. Clinical severity data and genotype correlation:
Did this study collect clinical data, especially the severity?
These data could provide useful correlations between genotype (e.g., the 111nt-dup variant) and clinical outcomes, potentially enhancing the relevance of genomic findings. If severity data were not collected, suggest mentioning this in the Limitations and suggest it as an area for future investigation.

5. Detection of genotype B:
The Discussion suggests that the low detection rate of genotype B may be due to experimental factors. This point should be elaborated, were the primers used for PCR or sequencing designed to target conserved regions across genotypes A and B? 
Is there any evidence of reduced sensitivity for detecting genotype B? 
Clarifying this will strengthen the interpretation of genotype distribution.

Typos.
1. Table 1: 
In the column labelled "x²", please consider replacing "x" with the proper Greek letter "χ" to accurately represent the Chi-squared (χ²) statistic.
